# Determination of the Pharmacokinetics and Pharmacodynamics of Isoniazid, Rifampicin, Pyrazinamide and Ethambutol in a Cross-Over Cynomolgus Macaque Model of *Mycobacterium tuberculosis* Infection

**DOI:** 10.3390/pharmaceutics14122666

**Published:** 2022-11-30

**Authors:** Laura Sibley, Andrew D. White, Charlotte Sarfas, Jennie Gullick, Fergus Gleeson, Faye Lanni, Simon Clark, Emma Rayner, Santiago Ferrer-Bazaga, Fatima Ortega-Muro, Laura Alameda, Joaquin Rullas, Veronica Sousa, Marisa Martinez, Inigo Angulo-Barturen, Adolfo Garcia, Juan José Vaquero, Henry E. Pertinez, Geraint Davies, Mike Dennis, Ann Williams, Sally Sharpe

**Affiliations:** 1UK Health Security Agency, Research and Evaluation, Porton Down, Salisbury SP4 0JG, UK; 2Department of Oncology, Medical Sciences Division, Churchill Hospital, Oxford OX3 7DQ, UK; 3Departmento de Bioingenieria, Universidad Carlos III de Madrid, 28903 Madrid, Spain; 4GlaxoSmithKlein, Research and Development, Diseases of the Developing World, Severo Ochoa, 2., 28760 Madrid, Spain; 5Department of Molecular and Clinical Pharmacology, University of Liverpool, Liverpool L69 3GX, UK

**Keywords:** tuberculosis, antibiotics, pharmacokinetics, pharmacodynamics, primates

## Abstract

Innovative cross-over study designs were explored in non-human primate (NHP) studies to determine the value of this approach for the evaluation of drug efficacy against tuberculosis (TB). Firstly, the pharmacokinetics (PK) of each of the drugs Isoniazid (H), Rifampicin (R), Pyrazinamide (Z) and Ethambutol (E), that are standardly used for the treatment of tuberculosis, was established in the blood of macaques after oral dosing as a monotherapy or in combination. Two studies were conducted to evaluate the pharmacokinetics and pharmacodynamics of different drug combinations using cross-over designs. The first employed a balanced, three-period Pigeon design with an extra period; this ensured that treatment by period interactions and carry-over could be detected comparing the treatments HR, HZ and HRZ using H37Rv as the challenge strain of *Mycobacterium tuberculosis* (M. tb). Although the design accounted for considerable variability between animals, the three regimens evaluated could not be distinguished using any of the alternative endpoints assessed. However, the degree of pathology achieved using H37Rv in the model during this study was less than expected. Based on these findings, a second experiment using a classical AB/BA design comparing HE with HRZ was conducted using the M. tb Erdman strain. More extensive pathology was observed, and differences in computerized tomography (CT) scores and bacteriology counts in the lungs were detected, although due to the small group sizes, clearer differences were not distinguished. Type 1 T helper (Th1) cell response profiles were characterized using the IFN-γ ELISPOT assay and revealed differences between drug treatments that corresponded to decreases in disease burden. Therefore, the studies performed support the utility of the NHP model for the determination of PK/PD of TB drugs, although further work is required to optimize the use of cross-over study designs.

## 1. Introduction

*Mycobacterium tuberculosis* (M. tb) is the causative agent of tuberculosis (TB) and was responsible for 1.3 million deaths in 2020 [1]. Effective drugs are important for the treatment of TB, and generally, treatment consists of a six-month regimen with the first-line drugs: Isoniazid, Rifampicin Ethambutol and Pyrazinamide, in different combinations to prevent the development of resistance and improve treatment success. The treatment success is reportedly 85% for those with drug-sensitive TB but treatment of rifampicin-resistant TB (RR-TB) and multidrug-resistant TB (MDR-TB) are less successful and more expensive (57%)(1). Furthermore, the prolonged treatment time and the side effects caused by the current drug regimens may lead to non-compliance that when coupled with issues with drug availability in low-income countries, can lead to an increase in drug-resistant TB. In 2020, 71% of bacteriology confirmed pulmonary TB cases had RR-TB, of which 132, 222 cases had MDR-TB, and 25,681 cases had XDR-TB [1).

To improve the prospect for drug therapies, novel drugs are being developed, as are strategies to reduce the length of time required for treatment. Pre-clinical in vivo models are critical to the evaluation pathway for new TB drugs. Mice are often used to provide an initial assessment of the pharmacokinetics and pharmacodynamics (PK/PD) and toxicity of drugs, but the difference in TB disease development in the mouse model can alter the PK/PD of the drugs. Consequently, PK/PD studies are also conducted in guinea pigs in which TB results in granulomas that mimic those in people, enabling the action of the drugs to be assessed in a more relevant model of human disease [2]. 

NHP models such as the macaque [3,4,5] or marmoset [6] more closely reproduce the classical pathology of tuberculosis in people, including cavitary lesions, but they require specialist expertise in the technical and welfare aspects of care and the highest level of ethical justification amongst animal models. Since endpoints based on destructive sampling are less feasible with these models and humane considerations place limitations on the intensity and invasiveness of samples, most of such studies are not based on the quantification of mycobacteria but rather measures of pathology or host response, particularly pulmonary imaging [4]. Studies using conventional parallel (AB) study designs for the evaluation of tuberculosis drug regimens have limitations which include treatment group inter-animal variability that can confound the interpretation of drug treatment effects without the use of unfeasibly large group sizes. By contrast, cross-over designs provide the opportunity to control for inter-animal variability as well as reducing the total number of animals required for each study. These designs have not previously been used in preclinical development in tuberculosis. The potential to reduce the number of animals required for drug evaluation studies would represent an important contribution to the 3Rs agenda in this area, where the aims are to Reduce the number of animals used for experimentation, Replace animals with alternatives including in vitro assays and modeling and Refine experiments to improve animal welfare and reduce suffering.

The work described here sets out to evaluate the PK/PD of the drugs Isoniazid, Rifampicin, Ethambutol and Pyrazinamide, which have previously been shown to successfully treat both natural [7] and experimental [4] TB infection in macaques. The actions of these drugs are suspected to be critically dependent on the presence of pathological changes similar to human disease. This can be clearly and reliably demonstrated in the preclinical studies prior to early phase clinical trials in humans and can be used to focus on the refinement of PK/PD studies using cross-over study designs. In order to carry out PK/PD studies in the macaque model, pilot PK studies were first performed to determine optimum doses of the anti-TB drugs. The PD and efficacy of Isoniazid, Rifampicin, Ethambutol and Pyrazinamide, delivered as combination therapies, were then evaluated in two studies using the cynomolgus macaque M. tb challenge model. The first study used a balanced Pigeon cross-over design to assess the PK/PD dynamics of Isoniazid, Rifampicin and Pyrazinamide combination therapies, and the second was refined to assess a balanced cross-over AB/BA design compared the impact of the two most potent sterilizing drugs (Rifampicin and Pyrazinamide) against the weakest (Ethambutol) in the current first-line regimen on a background of Isoniazid to prevent emergence of resistance.

Post-exposure vaccination offers the potential to shorten chemotherapy by enhancing the M. tb specific immune response [8,9], and understanding the impact of drug treatment on the response to infection is central to the success of such strategies. In this study, we employed the IFN-γ ELISPOT, a common immunoassay used to diagnose TB [10] and evaluate TB vaccine immune responses in humans and primates [11,12]; these data were then used to characterize the effect of the drug combinations on the Type 1 T helper (Th1) response to TB to understand whether they are beneficial or detrimental and perhaps direct how they could be improved.

An understanding of the PK/PD of anti-TB drug candidates will assist the development of new interventions, as the non-human primate provides the most relevant model of human tuberculosis. Our series of studies explored the feasibility and parameters of screening novel anti-TB compounds and regimens in the NHP model as part of pre-clinical development platforms.

## 2. Materials and Methods

### 2.1. Experimental Animals

Male cynomolgus macaques (*Macaca fascicularis*) aged 2.8–4.5 years old, were obtained for these studies from the breeding unit managed by the UK Health Security Agency (UK HSA). Genetic analysis of this colony has previously confirmed the cynomolgus macaques to be of Indonesian genotype [13]. Absence of previous exposure to mycobacterial antigens was confirmed by screening immediately prior to study for responses to PPD (SSI, Copenhagen, Denmark) and pooled 15-mer peptides of ESAT6 and CFP10 (Peptide Protein Research LTD, Fareham, UK) using an ex vivo IFN-γ ELISPOT assay (MabTech, Nacka, Sweden) as described elsewhere [12].

Animals were housed in compatible social groups, in accordance with the Home Office (UK) Code of Practice for the housing and Care of Animals Bred, Supplied or Used for Scientific Purposes, December 2014, and the National Committee for Refinement, Reduction and Replacement (NC3Rs), Guidelines on Primate Accommodation, Care and Use, August 2006 (NC3Rs, 2006). The housing environment was maintained within a temperature range of 18–24 °C and a relative humidity range of 40 to 70%. Cages were constructed with high-level observation balconies and enrichment was afforded by the provision of toys, swings, feeding puzzles and DVDs for multi-sense stimulation. Banks of cages were placed in directional airflow containment systems that allowed group housing and environmental control whilst providing a continuous, standardized inward flow of fully conditioned fresh air identical for all groups. In addition to standard Old-World primate pellets (Primate Mazuri Expanded MP(E), Dietex International, Witham, UK), a selection of differing vegetables and fruit was provided daily. Animals were sedated by intramuscular (IM) injection with ketamine hydrochloride (Ketaset, 100 mg/mL, Fort Dodge Animal Health Ltd., Southampton, UK; 10 mg/kg) for simple procedures such as blood sampling that required removal from their housing. None of the animals had been used previously for experimental procedures. All animal procedures and study design were approved by the Public Health England, Porton Down Establishment Animal Welfare and Ethical Review Body and authorized under an appropriate UK Home Office project license and as in previous studies [12].

### 2.2. Animal Study Designs

#### Satellite Pharmacokinetic Studies

Pilot pharmacokinetic studies were conducted in groups of two or three cynomolgus macaques to determine the drug concentrations in blood after oral administration of Rifampicin (R), Pyrazinamide (Z), isoniazid (H) and Ethambutol (E) either as single compounds or in combinations using the treatment regimens shown in Table 1, with the aim of confirming attainment of target exposures and to aid dose selection if required, especially in the absence of well characterized PK of TB drugs in non-human primates in the literature. Blood samples were collected prior to dosing at T = 0:00, and at three time points within a 30-min period of anesthesia on up to four occasions during the first 26.5 h after dosing (Table 1). Where possible, the aim would be to create composite pooled profiles across subjects (while acknowledging inter-individual variability) to overcome the limits of sparse sampling under anesthesia.

### 2.3. Drug Treatment Study 1: Pigeon Balanced Cross-Over Design with Extra Period Design

Nine cynomolgus macaques infected with M. tb H37Rv strain following exposure to aerosols estimated to contain presented doses of between 250 and 256 CFU were enrolled in a study to assess the efficacy of individual anti-TB therapy regimens. Drug treatment was initiated twelve weeks after infection with M. tb and proceeded through four phases of anti-TB combination therapy or control substance administration, as specified in the Balanced Pigeon cross-over study design [14] (Figure 1A). The three drug combinations assessed were: (i) Isoniazid (Sigma-Aldrich, St. Louis, MO, USA) dose 15 mg/kg, Rifampicin (Sigma-Aldrich) 15 mg/kg and Pyrazinamide (Sigma-Aldrich) 200 mg/kg; (ii) Isoniazid (Sigma-Aldrich) dose 15 mg/kg, Rifampicin (Sigma-Aldrich) 15 mg/kg; and (iii) Isoniazid (Sigma-Aldrich) dose 15 mg/kg, Pyrazinamide (Sigma-Aldrich) 200 mg/kg. The study design was fully balanced for first and higher order carry-over effects. During each treatment block, animals received a daily oral dose of the appropriate drug combination formulated in a four ml volume of fruit puree. At the end of the study schedule, animals continued to receive daily doses of the relevant test substance. Oral dosing ceased on the day prior to necropsy. Computed tomography (CT) scans were collected prior to the start of treatment at weeks three and 12 and at the end of each drug treatment phase at weeks 16, 20, 24 and 28. 

### 2.4. PK/PD Study 2—Two Period AB/BA Design

Eight cynomolgus macaques infected with M. tb Erdman strain following exposure to aerosols estimated to contain presented doses of between 192 and 271 CFU were enrolled in a study to assess the efficacy of two anti-TB therapy regimens. Drug treatment was initiated twelve weeks after infection with M. tb. The four macaques in group A received eight weeks of treatment with a combination of Isoniazid (Sigma-Aldrich) dose 15 mg/kg, Rifampicin (Sigma-Aldrich) 15 mg/kg and Pyrazinamide (Sigma-Aldrich) 200 mg/kg, followed by eight weeks of treatment with Isoniazid (Sigma-Aldrich) dose 15 mg/kg and Ethambutol (Sigma-Aldrich) 75 mg/kg (Figure 1B). The same anti-TB combination therapies were provided in the reverse order, using the same schedule to the four macaques in group B. During each treatment block, animals received a daily oral dose of the appropriate drug combination formulated in a 4 ml volume of fruit puree. At the end of the study schedule, animals continued to receive daily doses of the relevant test substance. Oral dosing ceased on the day prior to necropsy. CT scans were collected prior to the start of treatment at week eight, then at the mid and end of each drug treatment phase at weeks 12, 16, 20 and 24.

### 2.5. M. tb Challenge Strains

The M. tb H37Rv (National Collection of Type Cultures (NCTC) 7416) challenge stock was generated from a culture grown in a chemostat to steady state under controlled conditions at 37 °C ± 0.1, pH 7.0 ± 0.1 and a dissolved oxygen tension of 10% ± 0.1, in a defined medium, the details of which have been previously described [15,16]. High-concentration culture (approximately 2 × 10^8^ CFU/mL^−1^) was aliquoted and frozen at −80 °C. Titer of the stock suspension was determined from thawed aliquots by enumeration of CFU cultured onto Middlebrook 7H11 supplemented with oleic acid, albumin, dextrose and catalase (OADC) selective agar (BioMerieux, Basingstoke, UK). Aliquots were stored at −80 °C. For challenge, three vials of the selected strain were thawed and diluted appropriately, in sterile distilled water.

The M. tb Erdman K01 stock (BEI Resources, HPA-Sept 2011) was used for challenge as previously described [17]. A stock suspension was initially prepared from a 5 mL starter culture originally generated from colonies grown on Middlebrook 7H11 selective agar supplemented with oleic acid, albumin, dextrose and catalase (OADC) (BioMerieux, Basingstoke, UK). A liquid batch culture was then grown to logarithmic phase in 7H9 medium (Sigma-Aldrich, St. Louis, MO, USA) supplemented with 0.05% (*v*/*v*) Tween 80 (Sigma-Aldrich, St. Louis, MO, USA). Aliquots were stored at −80 °C. Titer of the stock suspension was determined from thawed aliquots by the enumeration of colony-forming units (CFU) cultured onto Middlebrook 7H11 OADC selective agar. On the day of challenge, three vials were thawed, pooled and diluted appropriately in sterile distilled water.

### 2.6. Aerosol Exposure to M. tb

The methodology and apparatus used to deliver M. tb via the aerosol route was as previously described [18]. In brief, mono-dispersed bacteria in particles were generated using a 3-jet Collison nebulizer (BGI) in conjunction with a modified Henderson apparatus [19], delivered to the nares of each sedated primate via a modified veterinary anesthesia mask. Challenge was performed on sedated animals placed within a ‘head-out’, plethysmography chamber (Buxco, Wilmington, NC, USA) to enable the aerosol to be delivered simultaneously with the measurement of respired volume. The calculations to derive the presented dose (PD) and the retained dose (the number of organisms assumed to be retained in the lung) have been described previously for aerosol doses [16,20]. The challenge was conducted such that one animal from each group was exposed in sequence with the cycle repeated until all animals were exposed.

### 2.7. Clinical Assessment

Animals were monitored daily for behavioral abnormalities including depression, withdrawal from the group, aggression, changes in feeding patterns and abnormal clinical parameters including dyspnea, tachypnoea and the presence of a cough. Animals were weighed, examined for external abnormalities and rectal temperature measured on each occasion that required blood sample collection, aerosol challenge or euthanasia. Red blood cell (RBC) hemoglobin levels were measured using a HaemaCue haemoglobinometer (Haemacue Ltd., Dronfield, UK) to identify the presence of anemia, and erythrocyte sedimentation rates (ESR) were measured using the Sediplast system (Guest Medical, Edenbridge, UK) to detect and monitor inflammation induced by infection with M. tb.

### 2.8. Preparation and Administration of Test SUBSTANCES 

Combinations of Isoniazid, Rifampicin, Pyrazinamide and Ethambutol Dihydrochloride (All Sigma-Aldrich) were prepared according to study schedules. Drug doses were prepared specifically for each subject based on body weight, collected not more than three days before dose delivery. Each animal’s specific drug dose was weighed in its solid state and mixed with fruit puree, either directly or after being dissolved into solution using the diluents appropriate for each drug: Rifampicin: water containing 20% Encapsin (Sigma-Aldrich); Pyrazinamide: water containing 1% Methyl Cellulose (Sigma-Aldrich). Four milliliters of fruit puree (‘Ella’s Kitchen’, apple and banana, UK) was drawn up into a sterile 5 mL luer lock syringe; this was then transferred via a syringe connector to the separate 5 mL luer lock syringe containing the measured drug. A uniform suspension of drug in puree was obtained by gently passing the suspension between the two syringes, after which the drug formulation syringe was capped and stored at 4–8 °C until use. Each formulation was used within three days of preparation. During each treatment block, animals received a daily dose of the appropriate test substance/regimen delivered via the oral route. 

### 2.9. Collection of Blood Samples for PK Analysis during Treatment Phase

For all studies, 2 mL blood samples were collected from each animal immediately prior to the administration of treatment to provide a ‘pre-dose’ PK sample for evaluation of ‘trough’ drug level during the study. For the PK studies, PK sampling was performed in all animals on a single occasion during each of the first two study periods on active treatment. PK sampling used a D-optimal sparse design consisting of two samples during not more than two periods of anesthesia per animal on each occasion. This approach resulted in 24 samples from a total of six animals on three separate occasions during the study for each drug assessed. During the PD studies, a 2 mL blood sample was collected by venipuncture via the femoral vein, for PK analysis at either 3.15 h or 6 h following administration of treatment on alternate occasions, as determined by the initial PK studies to be the optimal time points to measure drug availability. A blood sample to provide a PK trough sample was collected on the day of necropsy 24 h after the final drug administration. Blood samples were mixed with double the volume of MilliQ water, snap frozen and then stored at −80 °C until the time of analysis. For study 1, PK samples were collected at week 13 and at two weekly intervals until the end of the study. For study 2, PK samples were collected at two weekly intervals from week 8.

### 2.10. Samples Processing and LC-MS/MS Methods 

Primary stock solutions of 1 mg/mL of Rifampicin, Isoniazid or Ethambutol, and 3 mg/mL of Pyrazinamide, were prepared in Methanol:Water (1:1). In order to prepare calibration standards, working stock solutions in Methanol:Water (1:1) were prepared from the primary stocks. 5 uL of the corresponding working solution were added to 95 uL of blood to give final calibration standards. Additionally, 200 uL of MilliQ water were added to the blood samples to lyse the blood cells. Acetonitrile:Methanol (80:20) with Reserpine and Midazolam as internal standards was prepared as protein precipitant. BHT (butylhydroxytoluene) was added as antioxidant agent at 1% *w*/*v* concentration. Then, 190 uL of the protein precipitation solvent was added to each well of a filter plate 96-well, and 15 uL of sample (blank, unknown sample or standard curve) were added to each well. The samples were mixed and filtered. Then, samples were evaporated by ultravap evaporation at 37 °C for 15 min and reconstituted by adding 100 uL of 0.1% HFBA with 15 mg/mL of ascorbic acid. LC-MS/MS analysis was performed on Acquity UPLC (Waters)-Quattro Premier (Waters) to quantify each drug in blood. Chromatography was performed on Acquity UPLC BEH C18 50 × 2.1 mm 1.7 um column (Waters) using a reverse phase gradient. Then, 0.1% HFBA + 0.1% Acetic Acid in water was used for the aqueous mobile phase and Acetonitrile for the organic mobile phase. Multiple-reaction monitoring of parent/daughter transitions in electrospray positive-ionization mode was used to quantify the analytes. The following MRM transitions were used for Isoniazid (138.02/120.98), Rifampicin (823.55/791.2), Pyrazinamide (124.02/78.69) and Ethambutol (205.24/115.99).

### 2.11. PK Analysis 

Blood concentration–time profiles for all drugs from all treatment regimens were fitted with a one-compartment disposition PK model with first-order absorption input, using the R data analysis environment (v 3.4) [21], and treating the composite PK profiles from all subjects for a given treatment as a naïve pool. Fittings made use of the Pracma library and lsqnonlin function for nonlinear regression [22]. The mean maximum concentrations (Cmax) and times of maximum concentration (Tmax) were calculated across all individual subjects.

### 2.12. Computed Tomography (CT) Imaging 

CT scans were collected from sedated animals using a 16 slice Lightspeed CT scanner (General Electric Healthcare, Milwaukee, WI, USA) as described previously [3]. To improve the characterization of lesions and lymph nodes, such as the detection of necrosis, a non-ionic, iodinated contrast medium (Niopam 300 (Bracco, Milan, Italy)) was administered intravenously (IV) through the saphenous vein at 2 mL/kg body weight. Scans were evaluated by an expert radiologist with 30 years’ experience of interpreting chest CT scans blinded to the animal’s treatment and clinical status, for the number and distribution of pulmonary lesions across lung lobes) and the presence of nodules, cavitation, conglomeration, consolidation as an indicator of alveolar pneumonia, a ‘Tree-in–bud’ pattern as an indicator of bronchocentric pneumonia and lobular collapse as described elsewhere [23]. The airways were evaluated for the occurrence of wall thickening and the presence of bronchocoeles. The lymph nodes were assessed for enlargement and the presence of necrosis. Extra-pulmonary tissues, namely the liver, kidneys, and spleen were examined for the presence of single or multiple foci of disease, cavitation or necrosis. The pulmonary disease burden attributable to infection with *M. tuberculosis* was scored using a relative scoring system based on the number of lesions and the presence and extent of TB-induced structural abnormalities in the lungs as described previously [23]. The lungs were segmented using two procedures for volumetric CT assessment: (1) Manual lung segmentation using the tools available in the LUNGSIMI software; (2) Automatic lung segmentation (Hu) and airway extraction (Artaechevarria) using in-house developed software. Lung tissues were classified as ‘healthy’ or ‘sick’, with ‘sick’ parenchyma further divided as ‘soft’ or ‘hard’ tissue depending on the Hounsfield units (HU) values as reported in Chen et al. [23]. The thresholds (−1024, −575, −362) were adjusted by visual inspection, and the 63 scans were evaluated with the same thresholds. 

### 2.13. Necropsy

Animals were anaesthetized, and clinical data were collected. Blood samples were collected prior to euthanasia by intra-cardiac injection of a lethal dose of anesthetic (Dolelethal, Vétoquinol UK Ltd., 140 mg/kg, Towcester, UK). A post-mortem examination was performed immediately, and gross pathological changes were scored using an established system based on the number and extent of lesions present in the lungs, spleen, liver, kidney and lymph nodes, as described previously [18]. Samples of spleen, liver, kidneys and hilar lymph nodes were removed and sampled for quantitative bacteriology. The lungs, including the heart and attached tracheobronchial and associated lymph nodes, were removed intact and examined for lesions. The complete lung set was fixed by intra-tracheal infusion with 10% neutral buffered formalin (NBF) using a syringe and 13CH Nelaton catheter (J.A.K. Marketing, York, UK). The catheter tip was inserted into each bronchus in turn via the trachea; the lungs were infused until they were expanded to a size considered to be normal inspiratory dimensions, and the trachea was ligated to retain the fluid. The infused lung was immersed in 10% NBF. In addition, samples of kidneys, liver, spleen, and sub clavicular, hepatic inguinal and axillary lymph nodes were fixed in 10% NBF.

### 2.14. Histopathological Examination

Representative samples from each lung lobe and other organs were processed to paraffin wax, sectioned at 3–5 µm and stained with hematoxylin and eosin (HE). For each lung lobe, tissue slices containing obvious lesions were selected for histological examination. Where gross lesions were not visible, a sample was taken from a pre-defined anatomical location from each lobe to establish consistency between animals. Sections of lung associated lymph nodes (trachea–bronchial at the bifurcation and cranial and caudal to the bifurcation) and other tissues were evaluated for the presence of tuberculous lesions. Lesions were classified according to the scheme described previously [24]. In brief; type 1—small diffuse foci of macrophages and lymphocytes lacking clear boundaries; type 2—larger unorganized, larger, circumscribed, variably demarcated boundaries; type 3—granulomas same as type 2 but with central necrosis characterized by nuclear pyknosis and karyorrhexis; type 4—granulomas with evidence of organization of lymphocytes to a peripheral location; type 5—same as type 4 with central necrosis; type 6—classic, well-demarcated lesion with central, caseated necrosis and a variable rim of lymphocytes.

### 2.15. Bacteriology

The spleen, kidneys, liver and tracheobronchial lymph nodes were sampled for the presence of viable M. tb post-mortem as described previously(18). Weighed tissue samples were homogenized in 2 mL (spleen, liver, kidney, hilar lymph nodes) or 10 mL (lung lobes) of sterile water; then, they were either serially diluted in sterile water prior to being plated or plated directly onto Middlebrook 7H11 OADC selective agar. Plates were incubated for three weeks at 37 °C, and resultant colonies were counted.

### 2.16. Peripheral Blood Mononuclear Cells (PBMC) Preparation

Peripheral blood mononuclear cells (PBMC) were isolated from heparin anti-coagulated blood using standard methods [25].

### 2.17. Interferon-Gamma (IFN-γ) ELISPOT 

An IFN-γ ELISPOT assay was used to quantify the number mycobacteria-specific IFN-γ producing T cells in PBMCs using a human/simian IFN-γ kit (MabTech, Nacka. Sweden), as described previously [26]. PBMCs were cultured with 10 µg/mL PPD (SSI, Copenhagen, Denmark), or a pool containing overlapping 15 mer peptides spanning ESAT6 (Peptide Protein Research Ltd., Wickham, UK), or without antigen, in triplicate, and incubated for 18 h. Phorbol 12-myristate (Sigma-Aldrich Dorset, UK) (100 ng/mL) and ionomycin (CN Biosciences, Nottingham, UK) (1 µg/mL) were used as a positive control. After culture, spots were developed according to the manufacturer’s instructions. Plates were scanned and spots were enumerated using a CTL Immunospot S6 reader and software. Determinations from replicate tests were averaged, and data were analyzed by subtracting the mean number of spots in the medium-only control wells from the mean counts of spots in wells with antigen, or peptide pools, to derive an antigen-specific spot count. This value was multiplied by a factor of five and reported as an IFN-γ spot forming unit (SFU) frequency per million PBMCs.

### 2.18. Statistical Analysis

GraphPad Prism v7.0 and v8.0 (Graphpad Inc., San Diego, CA, USA) was used to construct graphs and analyze medians and areas under the curves. Group medians were analyzed using Mann–Whitney U tests. Spearman Rank Correlations were used to interrogate for relationships between pathology scores and M. tb antigen-specific IFN-γ producing cell frequencies.

Antigen-specific IFN-γ SFU profiles were plotted using Graphpad v7.0 (Graphpad Inc., San Diego, CA, USA) and used to calculate area under the curve (AUC) values for comparison of treatment group median AUC by Mann–Whitney U-test.

Fixed and random-effects analysis of variance accounting for multiple sources of variability including inter-individual, period, treatment and treatment-by-period interaction effects was performed in R 3.4 using the functions lm and lme. Models were compared on the basis of the AIC, Likelihood Ratio test and model assumptions checked using standard regression diagnostic plots.

## 3. Results

### 3.1. Satellite Pharmacokinetic Studies

Summary PK parameters from pilot PK studies are presented in Appendix A, including parameters from one-compartment 1st order absorption PK model fittings to naïve pooled data. 

Pilot PK study 1 measured drug concentrations in peripheral blood following oral administration of Rifampicin and Pyrazinamide at weight normalized doses matched to those used in humans of 10 mg/kg Rifampicin and 30 mg/kg Pyrazinamide given either singly or in combination (Figure 2). At 10 mg/kg, the observed blood AUC and Cmax for Rifampicin were 10.5 mg.h/L and 1460 ng/mL, respectively, when dosed alone and 31.7 mg.h/L and 2658 ng/mL when co-administered with Pyrazinamide. The latter values for Rifampicin following co-administration are however skewed by higher exposure seen in one subject out of three, and in general, a significant drug interaction is not seen for Rifampicin when the two drugs were co-dosed. Pyrazinamide at 30 mg/kg showed an AUC and Cmax of 27 mg.h/L and 7803 ng/mL when dosed alone with similar values of 24.5 mg.h/L and 7317 ng/mL when co-administered with Rifampicin. For both drugs, these exposures were lower than those seen in humans for the weight normalized doses given. In this first study, the drugs were delivered in solution (dissolved in encapsin (Rif) or methyl cellulose (Pyr)) and mixed with fruit puree to mask taste on dosing. To enable assessment of higher doses in subsequent studies, unsolubilized drug powders were mixed with fruit puree for dosing as the doses required exceeded the solubility limits of these formulations. 

PK study 2 (Figure 2) sought to evaluate exposures at higher doses of 75 and 150 mg/kg Rifampicin and 400 and 600 mg/kg Pyrazinamide (dosed singly) to assess tolerance of higher doses, mitigate uncertainty in the changed dosing format and observe if closer exposures to those required in human for efficacy were achievable. Unfortunately, the higher dose Rifampicin profiles were different to that at 10 mg/kg with the elimination phase not characterized under the limited sampling scheme achievable. Mean Cmax at 75 and 150 mg/kg was 23,267 ng/mL and 38,700 ng/mL, respectively, which are nonlinear with respect to dose compared to Cmax seen at 10 mg/kg. It appears that at the higher doses, and possibly due to the different dosage form, saturation of Rifampicin absorption and prolonged slow release/absorption from a depot in the gut has occurred, leading to “flip-flop” pharmacokinetics combined with some potential saturation of clearance leading to higher Cmax than expected. Pyrazinamide also showed potential saturation of both absorption and clearance at the higher doses and changed dose form, with AUC and Cmax of 850 mg.h/L and 239,500 ng/mL, respectively, at 400 mg/kg dose, and 2060 mg.h/L and 249,500 ng/mL at 600 mg/kg. 

To confirm the optimal doses of Rifampicin and Pyrazinamide, a third study was conducted to test Rifampicin at 10 and 30 mg/kg to mitigate the risk of variability in its exposure and straddle the hypothesized optimum doses of 20 mg/kg and Pyrazinamide at 250 mg/kg that would achieve exposures known to be efficacious in human (Figure 2). Isoniazid was included in the combination treatment and a fourth block of sampling under anesthesia for evaluation of a late terminal phase PK sample period added to the study. Target exposures for the three drugs were met (Table 2), enabling the dose levels to be set for the PD studies: Isoniazid at 15 mg/kg, Rifampicin at 15 mg/kg and Pyrazinamide at 200 mg/kg.

A fourth PK study (Figure 2) was conducted to compare Ethambutol doses of 25 mg/kg and 75 mg/Kg. The 75 mg/kg dose resulted in a blood AUC of 47.1 mg.h/L which was in line with AUC quoted for Ethambutol plasma levels in humans, and 75 mg/kg would therefore be used as the dose for infection studies. PK exposure resulting from the 25 mg/kg dose showed greater variability with the calculated AUC to be 38 mg.h/L, which was higher than what would be expected for dose linearity.

### 3.2. PK/PD Study 1: Pigeon Balanced Cross over Design with Extra Period Design

The first cross-over study was constructed around a three-period Pigeon design with nine animals and treatment periods of four weeks with imaging performed at the end of each phase. Tuberculosis infection was initiated in all nine macaques following exposure to aerosols containing a median presented dose of 262 CFU (range 250—269 CFU) M. tb H37Rv providing an estimated median retained dose in the lung of 37 CFU. The impact of infection on the clinical condition of individuals was evaluated through the monitoring of body weight and temperature, ESR, red cell hemoglobin concentration and the collection of CT scans to visualize and quantify pulmonary disease (Figure 3A and Appendix A). Three weeks after exposure to M. tb, CT scans revealed discrete lesions in all animals distributed through all lung lobes which subsequently increased between week 3 and 12 (Figure 3B). 

Twelve weeks after M. tb challenge, animals were randomly assigned to one of nine sequences of four treatments (No treatment, Isoniazid and Rifampicin (HR), Isoniazid and Pyrazinamide (HZ) and Isoniazid, Rifampicin and Pyrazinamide (HRZ)). During the treatment phases, body weights generally remained stable or increased and nodule numbers and lung CT score decreased in all animals except 109HMD (Figure 3A–C). The volume of disease as measured by CT varied between animals and treatment combinations (Figure 3D). Enlarged necrotic lymph nodes were detected from week 12 but resolved by week 28. Pneumonia was observed from week 18 in two of the nine animals. At week 20–24, most of the animals had pneumonia, but this had resolved by week 28 for all animals (Figure 3E). 

At the end of the study, disease burden was evaluated using a gross pathology score system described previously [18] which showed scores to be similar between the treatment groups (Figure 3F), although the pair that received HZ during the final treatment phase both showed high pathology scores, whereas more variation was seen between pairs that received similar treatments. Bacterial burden was quantified in samples of spleen, liver, kidneys and hilar lymph nodes by culture on agar plates, which confirmed M. tb was present in all animals (Figure 3G) and predominately detected in the hilar lymph nodes (seven of nine). 

Histological assessment of the lung tissue scored the granuloma types based on an established scoring system (Figure 3H). Whilst results varied among the study subjects, those that ended the study on treatment with HR possessed a higher number of type 1 granulomas, which are small, lack clear boundaries and suggest disease control in line with the decrease in nodule counts identified from the CT scans. Although still present, disease appeared controlled as the volume of infected tissue plateaued. Conversely, those that ended on a phase without treatment showed a higher proportion of type 5 lesions with central necrosis that suggests the disease was more severe in these animals. 

Evaluation of drug concentrations in blood samples collected in the six-hour period following drug dosing confirmed target exposures were achieved for all three drugs (Figure 3H).

#### 3.2.1. Treatment Effect Analysis

Analysis of the body weight and imaging data using a fixed effects analysis of variance approach are summarized in Table 2. As expected, inter-animal variability was high for all the endpoints examined, and this was successfully removed by the cross-over design, which was represented by a highly statistically significant term in the model. For imaging measures, but not weight, there was also a highly significant period effect as disease improved during the study. However, the effect of treatment was not statistically significant overall, although the reduction in nodule count due to HR treatment approached statistical significance. Treatment effects were all smaller in size than the period effect (for example, maximum period effect was −10.1 nodules whereas the largest effect of treatment was −4.6 nodules). Present CT multiparametric analysis would help future experimental designs to identify, among the various lung changes seen in CT, those more informative of active tuberculosis pathology evolution

#### 3.2.2. Immune Response during Infection and Treatment

The IFN-γ response to TB antigens; PPD, ESAT6 and CFP10 was measured using an ex vivo ELISPOT assay applied at two-week intervals through the study (Figure 4A–C). Increased frequencies of PPD, CFP10 and ESAT-6-specific IFN-γ Spot-Forming Units (SFU) relative to pre-challenge levels were detected in all study subjects during the twelve weeks after challenge before treatment began. Typically, responses were detected from four weeks after infection with peak responses particularly those to PPD peaking at week four or six. Comparison of responses at the beginning and end of each treatment phase revealed PPD-specific cell frequencies decreased following treatment with HZ or no treatment, whereas a slight increase was observed during phases of treatment with HR (Figure 4D). ESAT6-specific responses increased when HRZ was given, decreased following HZ treatment and remained constant during HR or no treatment (Figure 4D). CFP-10-specific responses decreased following HZ and HRZ but remained stable after HR or no treatment (Figure 4D).

#### 3.2.3. PK/PD Study 2: Two Period AB/BA Design

The second study used a classical two period AB/BA design to compare treatment with Isoniazid, Rifampicin and Pyrazinamide (HRZ) or Isoniazid and Ethambutol (HE). Eight cynomolgus macaques were infected with M. tb Erdman strain following exposure to aerosols estimated to contain presented doses of between192 and 271 CFU. The clinical condition of individuals was evaluated throughout the study period by monitoring body weight and temperature, ESR, red cell hemoglobin concentration and CT scanning to visualize and quantify pulmonary disease (Figure 5A–E, Appendix A). 

Evaluation of body weight during treatment revealed a trend for subjects to gain weight during treatment with HRZ (Figure 5A). Over the course of both treatment phases, nodule numbers decreased in all animals; although those in group 1 (HE/HRZ) possessed lower numbers of nodules and volume of disease than group 2 (HRZ/HE), the differences did not reach significance (nodule counts *p* = 0.2286, volume of disease *p* = 0.8857) (Figure 5B,C). 

Before treatment, discrete nodules distributed though all lung lobes were detected in all eight animals with conglomerate nodules present in three animals three weeks after aerosol exposure to M. tb Erdman (Figure 5B). At week eight, nodule numbers had increased and conglomerate nodules were identified in six animals. The lung score and volume of disease was measured by CT increased in all animals and during phase 2; a trend for group 1 (HE/HRZ) to show lower lung scores and volume of disease in comparison to that in group 2 (HRZ/HE) was observed (Figure 5C,D).

Cavitated nodules were not seen in any of the animals before the start of treatment. Pneumonia was observed in four animals at week three and in five animals at week eight after infection. Lung-associated lymph nodes were enlarged in five animals, and of these, the nodes in three were also necrotic three weeks after infection and five weeks later, lymph nodes that were both enlarged and necrotic were observed in seven of the eight animals. Treatment commenced eight weeks after challenge, with subjects randomly assigned to the two treatment sequences. Cavitation and conglomeration were seen consistently throughout the treatment phase in a proportion of the animals (Figure 5E). In contrast, the number of animals showing enlarged, necrotic lymph nodes reduced from six to one over the course of treatment. By the end of the treatments at week 24, only one animal still had lymph node abnormalities, and the nodule counts were decreased in comparison to previous time points.

At the end of the study, disease burden was measured using a gross pathology scoring system which showed the levels of disease to be similar in the two study groups (Figure 5E). A significantly higher number of M. tb were isolated from the tissues collected from the group that received treatment with HE then HRZ than those that received the treatments in the reverse order (Figure 5F). Gross lesion counts were similar between the groups overall (Figure 5G). Microscopic examination of the lesion identified in lung sections to categorize the types of lesions (1-5; 1-3 being organized and 4-5 being disorganized) present showed that granuloma types were similar between the two groups, with mainly type 1 and type 5 granulomas being present, but a trend for more type 4 granulomas in group 2 (HRZ/HE). 

Evaluation of drug concentrations in blood samples collected during the six-hour period following drug dosing confirmed target exposures were achieved for all four drugs. Overall, for Isoniazid, Rifampicin and Pyrazinamide, detectable drug in the blood decreased between 3.15 and 6 h after dosing, (Figure 5H), whereas Ethambutol was detected at similar levels.

#### 3.2.4. Treatment Effect Analysis

Analysis of the body weight and imaging data sets using a fixed effects analysis of variance approach is summarized in Table 3, which confirmed strong subject (*p* = 0.018) and period (*p* = 0.023) effects and due to limitation of degrees of freedom for the classical primary analysis, a random-effects approach was adopted, since the estimation of individual subject effects was not of direct interest. The random effects for the subject were relatively large at approximately 50% of the mean fixed effect outcomes, and the period effect remained consistently significant in the random effects model. Despite effective adjustment for the period effects and accounting for within-subject correlation with baseline values within each period, however, no statistically significant difference was detected between HE and HRZ on any of the endpoints studied.

#### 3.2.5. Immune Response during Infection and Treatment

The frequencies of IFN-γ-producing cells specific for TB antigens PPD, ESAT6 and CFP10 were measured following infection and during drug therapy using an ex vivo ELISPOT assay applied at two-week intervals throughout the study (Figure 6A–C). Increased frequencies of PPD, CFP10 and ESAT-6-specific IFN-γ SFU relative to pre-challenge levels were detected in all study subjects during the eight weeks after challenge before treatment began (Figure 6A–C). During the first treatment phase, frequencies of PPD, CFP10 and ESAT6-specific IFN-γ secreting cells were higher in group 1 (HE/HRZ) relative to levels in group 2 (HRZ/HE) (Figure 6A–C), whereas the responses made by the two groups were similar during the second treatment phase. A reduction in CFP10 specific responses was noted at the end of the study (Figure 6C). The fold change in SFU/million cells between the level at the start and the level at the end of each treatment phase was determined in order to compare the responses made during the treatment phases and revealed a trend for PPD-specific SFU frequencies to decrease less during the HRZ treatment phase than during the HE phase (Figure 6D). Responses to ESAT6 and CFP10, which are associated with bacterial burden, decreased in both treatment phases but were more consistent during the HE phase. Increased frequencies of PPD-specific SFU measured as AUC correlated with a lower lung pathology score (r = 0.9011, *p* = 0.0018) (Appendix A). 

## 4. Discussion

This report describes the application of novel PK and PK/PD study designs in the macaque model to determine their suitability for the evaluation of drugs to combat tuberculosis. 

Satellite pharmacokinetic experiments were performed to clarify the decision-making processes in designing the efficacy studies such as choices in dosing levels, sampling time points, with particular emphasis on correctly setting the dose to achieve NHP exposures similar to those seen in humans for the TB drugs under examination rather than provide a statistically definitive account of NHP PK of the drugs and combinations used. The novel approach employed for sample collection that used overlapping time courses in individuals successfully provided a richer composite time course for evaluation. As the doses of drugs required exceeded the solubility limits of the formulation required, insolubilized drug powders were mixed in fruit puree to mask the taste during oral dosing which proved an effective approach at the dose levels used; however, higher dosing levels could potentially lead to depot formation in the intestinal tract, allowing prolonged absorption and resulting in increased apparent half-life. 

The first cross-over study was constructed around a three-period Pigeon design with nine animals and periods of four weeks with imaging performed at the end of each. The design was balanced to ensure estimation of an expected period effect and treatment-by-period interaction, and a final extra period was included to enable separate estimation of carry-over. This design addressed two key questions: first, could any of the treatments be distinguished from no treatment; and second, could the independent and potentially synergistic effects of R and Z be reproduced in a model with high biological validity? The cross-over design was logistically feasible, with all animals completing the study successfully and removed substantial inter-animal variability. A significant period effect was observed with a tendency for the number of lung nodules to decrease steadily over the periods. The treatment effects observed were smaller than either of these, although the reduction in nodule count due to HR approached statistical significance. Treatment-by-period interactions or carryover effects were not observed, and it is likely that the period trend relates to the development of immunity rather than pharmacokinetic or pharmacodynamic carry-over. The limited volume of disease induced by H37Rv may have reduced the ability of this approach to identify treatment effects which could be relatively larger with a greater volume of disease. For the second PK/PD study, the more virulent TB strain Erdman [27] was used with the aim was to increase pulmonary disease burden induced prior to the start of treatment. The duration of each treatment block was expanded from four to eight weeks to increase the size of treatment effect, and a simpler and shorter study design was used. To improve analysis of covariance, there was the inclusion of an additional midpoint scan—three CT scans per period, collected at the start, midpoint, and end of each treatment. These changes to the study design meant that differences were observed in terms of bacteriology read out and weight changes, and the simpler design was easier to analyze and consider carry-over effects between different drug treatment phases. It should be considered that CT scans are detecting tissue lesions that in most cases would incidentally resolve requiring additional time once the infection has been controlled. This feature must be taken into account and could be improved by adding functional imaging readouts (i.e., PET/CT) or tracers that could mark the bacteria directly.

Evaluation of the TB-specific IFN-γ-producing cell profiles specific for PPD, CFP10 and ESAT6 antigens in the macaques enrolled in study 2 revealed frequencies prior to treatment similar to previously published studies where cynomolgus macaques had been infected with the Erdman strain of M. tb [28,29] and act as confirmation that all animals were infected and initiated an immune response to infection. 

AUC analysis of the same pre-treatment period showed the frequency of ESAT6-specific SFU measured in the macaques in study 2 (Erdmann) to be four times greater than that measured in the macaques in study 1 (H37Rv). The magnitude of the ESAT6-specific IFN-γ response has been shown to be positively associated with bacterial burden [12,18,30]; therefore, the variation seen is likely due to the difference in the strains of M. tb used between the two studies. This could be due to the lower level of disease induced following infection with the H37Rv strain of M. tb observed in comparison to Erdman, which is in line with other published observations [27]. 

In both studies, the frequency of ESAT-6 specific SFU decreased during treatment, which suggests that treatment was successful and antigen load reduced due to treatment. During Study 1, the frequencies of ESAT-6 specific SFU were generally seen to increase during the periods when treatment was not given, suggesting unimpeded disease progression. Looking in more detail at the fold change in ESAT-6 specific SFU during specific treatment phases in Study 1, the strongest reduction was observed following HZ treatment in comparison to HR and HRZ. The addition of R to the treatment regimen appeared to affect the responses measured to ESAT-6 and may suggest a reduction in the efficacy of treatment, the cause of which would be important to dissect in future work. Overall, in Study 2, lower frequencies of ESAT-6 specific SFU were detected following HRZ treatment than after treatment with HE, suggesting the triple combination therapy to be more effective at reducing antigen load. However, the effect was not consistent, and it may be dependent on the timing and treatment phase in which the drug combination was given. By contrast, a comparison of responses measured at the beginning and end of each treatment phase revealed PPD-specific cell frequencies decreased most following either treatment with HZ or no treatment, whereas a smaller decrease was observed during phases of treatment with HR and HRZ. Similarly, there was a trend for PPD-specific SFU frequencies to decrease less during the HRZ treatment phase than during the HE phase in study 2. These observations may suggest a potential role for the frequency of PPD-specific IFN-γ secreting cells as a biomarker of successful treatment; Although a review conducted on the use of IFN-γ release assays (IGRA) to monitor infection during chemotherapy previously found variability to preclude use as a biomarker of successful treatment [31], perhaps the assay could be refined and utility improved. 

The immune response profiles measured by ELISPOT suggested that the drugs used in the study influenced the frequency of cells capable of producing cytokines in the peripheral blood. Chemoattractants such as IP-10 and MCP-1 have been reported to be up-regulated by TB treatments in mice, and pro-inflammatory cytokines including IFN-γ, TNFα, IL-12, IL-17 and IL-1β are associated with TB disease. Isoniazid has been demonstrated to cause apoptosis of Th1 cells, which is thought to lead to an increased likelihood of TB reactivation after the end of treatment [32]. Future work should evaluate the effect of drugs on the functionality of T-cells and identify whether this correlates to control of TB disease. The information gathered on the impact of front-line antibiotics on the immune response and particularly those specific for M. tb is critical to guide timings of post-exposure immunization schedules for TB and would assist decision making on the safety and efficacy of vaccination during drug treatment. The data would also aid future immunotherapy design to identify types of immune response to promote or avoid.

## 5. Conclusions

The experiments described here were designed to assess whether the NHP model could reproduce the strong effects of R and Z seen in human clinical trials and might therefore have a use in assessing the effect of drugs which may depend on specific pathological features similar to those found in people. Taken together, the results of the two studies do not suggest that this can be achieved using a cross-over approach, despite the high biological plausibility of the model in view of its ability to reproduce the features of human disease. Limitations of these experiments include the necessary small numbers of animals used, the strong period effect observed, which is probably due to acquired immunity in primary infection and which would likely also be observed in parallel group studies, and the use of structural but not functional information among the imaging endpoints. However, the NHP model may still have unique advantages in studies of PK-PD of drugs with pathology-dependent biological or pharmacological properties, and it is possible that refinements in imaging and biomarker technologies could address some of the limitations that we have identified even when efficient designs are used to effectively control the inter-individual and period variability that appear to be an irreducible feature of the model.

## Figures and Tables

**Figure 1 pharmaceutics-14-02666-f001:**
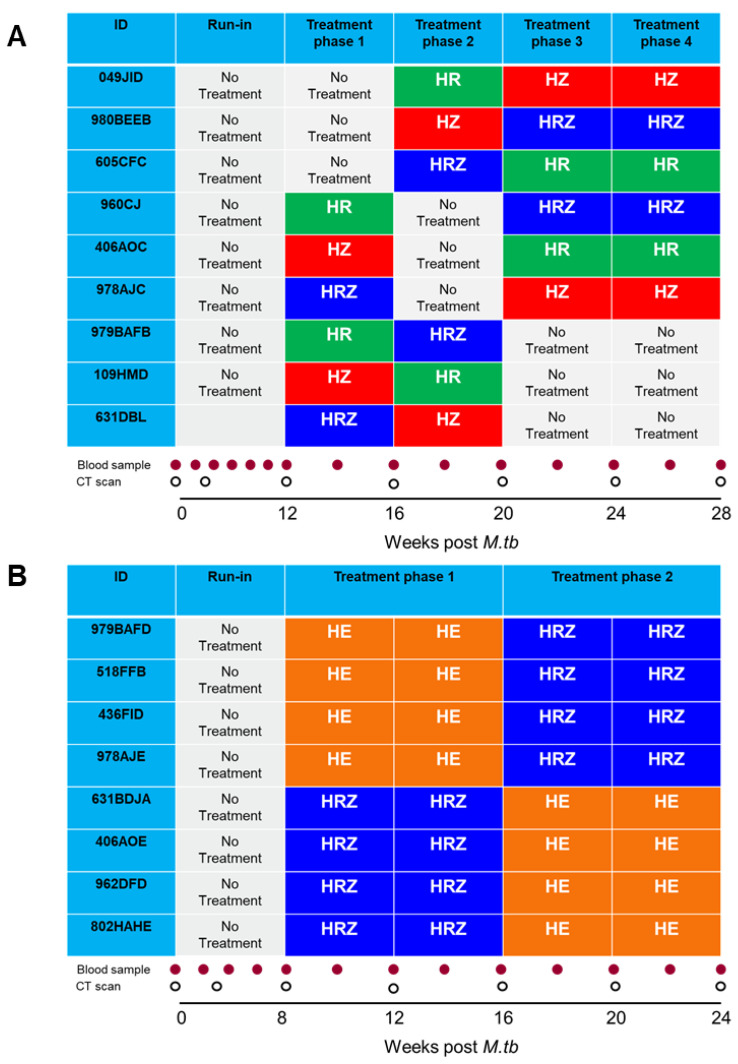
Study plans showing the treatment phases, blood sampling schedules and CT scan collection points used in PK/PD Study 1 (**A**) and 2 (**B**). HE (Isoniazid, Ethambutol); HRZ (Isoniazid, Rifampicin, Pyrazinamide); HZ (Isoniazid, Pyrazinamide); HR (Isoniazid, Pyrazinamide).

**Figure 2 pharmaceutics-14-02666-f002:**
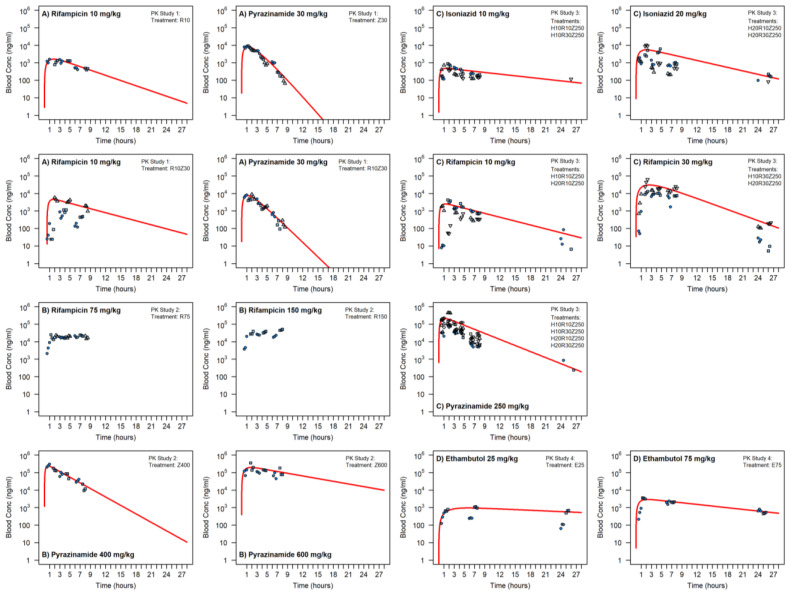
Blood PK of Rifampicin, Pyrazinamide Isoniazid and Ethambutol in cynomolgus macaques. Separate symbols for each individual subject, 1-compartment PK modeled fittings to naïve-pooled raw data (where feasible) shown by red line to derive CL, V, ka and AUC.

**Figure 3 pharmaceutics-14-02666-f003:**
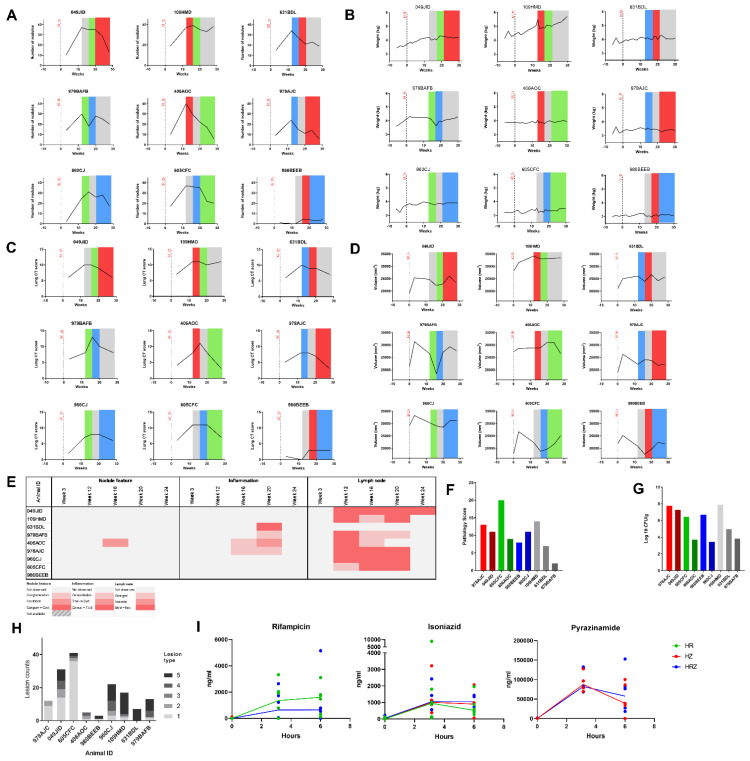
Measures of M. tb induced disease burden and drug exposure levels in blood from cynomolgus macaques following infection with M. tb H37Rv over three phases of treatment with combinations of HR, HZ or HRZ. (**A**) Pulmonary nodule counts measured from CT scans, (**B**) body weight profiles, (**C**) lung scores measured by CT, (**D**) volume of diseased lung measured from CT scans, (**E**) heat map of disease characteristics observed from CT scans, (**F**) gross pathology scores, (**G**) summed M. tb bacillary burden (CFU) cultured from extrapulmonary tissues, (**H**) pulmonary granuloma lesion numbers and types quantified during histopathology evaluation, (**I**) levels of Rifampicin, Isoniazid and Pyrazinamide detected in blood samples collected 3.15 and 6 h after administration, ng/mL. Red = HZ, green = HR, blue = HRZ, gray = no treatment.

**Figure 4 pharmaceutics-14-02666-f004:**
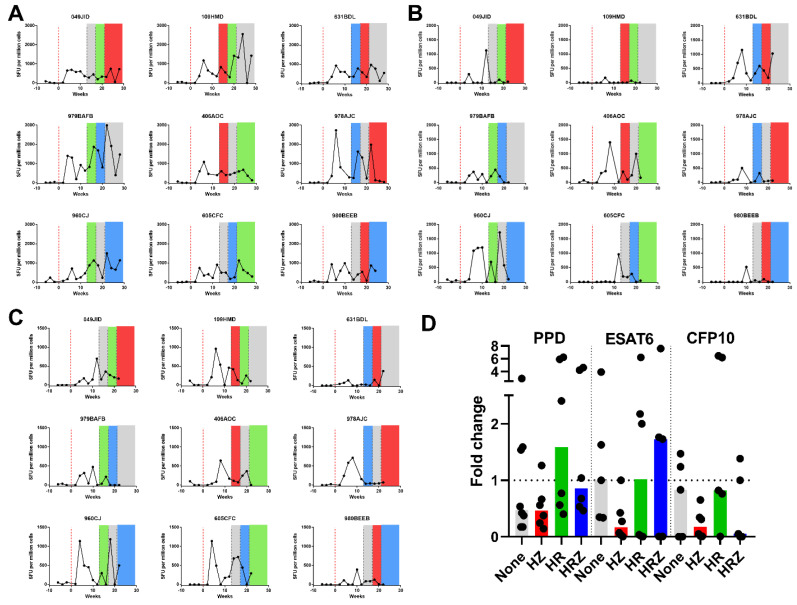
Frequencies of M. tb antigen specific IFN-γ producing SFU measured by ELISPOT in the peripheral blood from cynomolgus macaques infected with M. tb H37Rv during treatment with combinations of HR, HZ or HRZ: (**A**) PPD-specific SFU, (**B**) ESAT6-specific SFU, (**C**) CFP10-specific SFU, (**D**) Fold change in SFU/million cells between the start and end of treatment. Red = HZ, green = HR, blue = HRZ, gray = no treatment.

**Figure 5 pharmaceutics-14-02666-f005:**
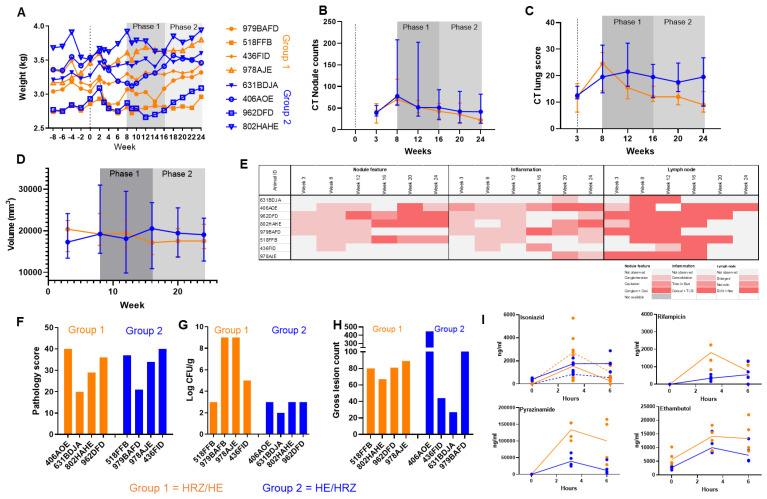
Measures of M. tb induced disease burden and drug exposure levels in blood in cynomolgus macaques following infection with M. tb Erdman and two phases of treatment with combinations of HRZ or HE. (**A**) Body weight profiles, (**B**) nodule counts measured from CT scans, (**C**) lung score measured by CT, (**D**) volume of diseased lung as measured by CT scan, (**E**) heatmap of disease characteristics as measured by CT scan, (**F**) gross pathology scores, (**G**) M. tb bacillary burden, (**H**) granuloma types assessed by histopathology by presence in different lung lobes, (**I**) levels of Rifampicin, Isoniazid, Pyrazinamide and Ethambutol detected in blood samples 3.15 and 6 h after administration. Orange = Group 1 (HRZ/HE), blue = Group 2 (HE/HRZ), Medians shown. ng/mL. Isoniazid was measured in both treatment phases so complete line = phase 1, dashed line = phase 2.

**Figure 6 pharmaceutics-14-02666-f006:**
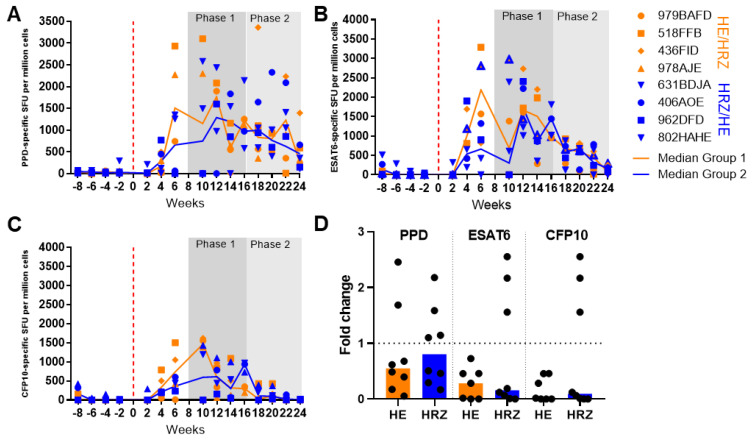
IFN-γ ELISPOT measurement of M. tb specific SFU in cynomolgus macaques infected with M. tb Erdman and following two treatment phases with different combinations of HRZ or HE (**A**) PPD-specific SFU, (**B**) ESAT6-specific SFU, (**C**) CFP10-specific SFU, (**D**) fold change SFU/million cells between the start and end of treatment phase, Orange = HE, blue = HRZ. Medians shown.

**Table 1 pharmaceutics-14-02666-t001:** Treatment dose and PK study blood sample collection schedule for assessment of drug pharmacokinetics in cynomolgus macaques.

Study	Treatment Dose (mg/Kg)	Drug Form	Subject Number	Sample Collection Time in Minutes Relative to Dosing												
				Pre	1st Sampling Period			2nd Sampling Period			3rd Sampling Period			4th Sampling Period		
	R 10	S	1	00:00	00:30	00:45	01:00	03:00	03:15	03:30	06:00	06:15	06:30	nd	nd	nd
			2	00:00	01:15	01:30	01:45	03:45	04:00	04:15	07:00	07:15	07:30	nd	nd	nd
			3	00:00	02:00	02:15	02:30	04:30	04:45	05:00	08:00	08:15	08:30	nd	nd	nd
1	Z 30	S	4	00:00	00:30	00:45	01:00	03:00	03:15	03:30	06:00	06:15	06:30	nd	nd	nd
			5	00:00	01:15	01:30	01:45	03:45	04:00	04:15	07:00	07:15	07:30	nd	nd	nd
			6	00:00	02:00	02:15	02:30	04:30	04:45	05:00	08:00	08:15	08:30	nd	nd	nd
	R 10, Z 30	S	7	00:00	00:30	00:45	01:00	03:00	03:15	03:30	06:00	06:15	06:30	nd	nd	nd
			8	00:00	01:15	01:30	01:45	03:45	04:00	04:15	07:00	07:15	07:30	nd	nd	nd
			9	00:00	02:00	02:15	02:30	04:30	04:45	05:00	08:00	08:15	08:30	nd	nd	nd
	R 75	D	10	00:00	00:30	00:45	01:00	03:00	03:15	03:30	06:00	06:15	06:30	nd	nd	nd
			11	00:00	01:15	01:30	01:45	03:45	04:00	04:15	07:00	07:15	07:30	nd	nd	nd
			12	00:00	02:00	02:15	02:30	04:30	04:45	05:00	08:00	08:15	08:30	nd	nd	nd
	R 150	D	13	00:00	00:30	00:45	01:00	03:00	03:15	03:30	06:15	06:30	06:45	nd	nd	nd
			14	00:00	01:45	02:00	02:15	04:15	04:30	04:45	07:30	07:45	08:00	nd	nd	nd
2	Z 400	D	15	00:00	00:30	00:45	01:00	03:00	03:15	03:30	06:15	06:30	06:45	nd	nd	nd
			16	00:00	01:45	02:00	02:15	04:15	04:30	04:45	07:30	07:45	08:00	nd	nd	nd
	Z 600	D	17	00:00	00:30	00:45	01:00	03:00	03:15	03:30	06:15	06:30	06:45	nd	nd	nd
			18	00:00	01:45	02:00	02:15	04:15	04:30	04:45	07:30	07:45	08:00	nd	nd	nd
	H 10; R 10, Z 250	D	19	00:00	00:30	00:45	01:00	03:00	03:15	03:30	06:15	06:30	06:45	24:00	24:15	24:30
			20	00:00	01:45	02:00	02:15	04:15	04:30	04:45	07:30	07:45	08:00	26:00	26:15	26:30
3	H 10; R 30, Z 250	D	21	00:00	00:30	00:45	01:00	03:00	03:15	03:30	06:15	06:30	06:45	24:00	24:15	24:30
			22	00:00	01:45	02:00	02:15	04:15	04:30	04:45	07:30	07:45	08:00	26:00	26:15	26:30
	H 20; R 10, Z 250	D	23	00:00	00:30	00:45	01:00	03:00	03:15	03:30	06:15	06:30	06:45	24:00	24:15	24:30
			24	00:00	01:45	02:00	02:15	04:15	04:30	04:45	07:30	07:45	08:00	26:00	26:15	26:30
	H 20; R 30, Z 250	D	25	00:00	00:30	00:45	01:00	03:00	03:15	03:30	06:15	06:30	06:45	24:00	24:15	24:30
			26	00:00	01:45	02:00	02:15	04:15	04:30	04:45	07:30	07:45	08:00	26:00	26:15	26:30
	E 25	D	27	00:00	00.30	00.45	01.00	06:00	06:15	06:30	24:00	24:15	24:30	nd	nd	nd
			28	00:00	01:15	01:30	01:45	07:00	07:15	07:30	25:00	25:15	25:30	nd	nd	nd
4	E 75	D	29	00:00	00.30	00.45	01.00	06:00	06:15	06:30	24:00	24:15	24:30	nd	nd	nd
			30	00:00	01:15	01:30	01:45	07:00	07:15	07:30	25:00	25:15	25:30	nd	nd	nd

R: Rifampicin; Z: Pyrazinamide; H: Isoniazid; E: Ethambutol; nd: not done, D: dry powder; S: solution.

**Table 2 pharmaceutics-14-02666-t002:** (A)—Akaike Information criterion for ANOVA models for study 1. (B)—Likelihood ratio test of ANOVA models of Study 1 data.

Model	AIC Of ANOVA Model Fit		
	Nodule Count	Weight	Vol_Disease
1. Base	276.5	67.5	744.1
2. ID_effect	245.2	−46.4	722.9
3. Period_effect	227.9	−49.8	715.3
4. Treatment_effect	225.1	−54.2	716.5
5. Treatment_period_int_effect	216.3	−57.4	692.1
6. Baseline_effect	226.6	−54.4	716.4
(A)			
**Test**	***p*-Value F-Test of Model Comparison**		
	**Nodule Count**	**Weight**	**Vol_Disease**
2 vs. 1: ID_effect	0.000	0.000	0.000
3 vs. 2: Period_effect	0.001	0.093	0.026
4 vs. 3: Treatment effect	0.153	0.103	0.416
5 vs. 4: Treatment_period_int_effect	0.261	0.446	0.040
6 vs. 4: Baseline effect	0.615	0.270	0.281
(B)			

**Table 3 pharmaceutics-14-02666-t003:** (A)—Akaike Information criterion for ANOVA models for study 2. (B)—Likelihood ratio test of ANOVA models of Study 2 data.

Model	AIC of ANOVA Model Fit		
	Nodule Count	Weight	Vol_Disease
1. base	33.18	7.78	−54.06
2. period	28.83	−0.783	−52.097
3. period + treatment	29.49	1.216	−51.719
4. treatment period interaction	30.995	2.795	−49.722
5. period + baseline	30.883	−11.97	−60.62
6. period + baseline + treatment	31.48	−9.972	−60.85
(A)			
**Test**	**F-Test of Model Comparison**		
	**Nodule Count**	**Weight**	**Vol_Disease**
2 vs. 1: period	0.012	0.001	0.042
3 vs. 2: period + treatment	0.247	0.973	0.203
4 vs. 2: treatment period interaction	0.399	0.421	0.443
5 vs. 2: period + baseline	0.97	0.0003	0.001
6. vs. 5: period + baseline + treatment	0.245	0.978	0.135
(B)			

## Data Availability

Data available on request from the corresponding author.

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
