# Peer review of "Determination of the Pharmacokinetics and Pharmacodynamics of Isoniazid, Rifampicin, Pyrazinamide and Ethambutol in a Cross-Over Cynomolgus Macaque Model of Mycobacterium tuberculosis Infection"

_pharmaceutics, 2022, doi:10.3390/pharmaceutics14122666_

Round 1
Reviewer 1 Report
This report describes the application of novel PK and PK/PD study designs in the macaque model to determine their suitability for the evaluation of drugs to combat tuberculosis. Model substances have been the drugs Isoniazid, Rifampicin, Ethambutol and Pyrazinamide, which have previously been shown to successfully treat both natural and experimental TB infection in macaques and are the standard drugs in the treatment of tuberculosis in man.
The aim of the study is to apply PK/PD models in non-human primates in the development of new anti-TB drug candidates to treat human tuberculosis.
Whereas the study design is optimal as well as the results, there is no discussion whether the model predicts the well-known pharmacokinetics and pharmacodynamics of the 4 standard drugs in humans. Without this the study remains a nice exercise in macaques and does not warrant publication.
Author Response
This report describes the application of novel PK and PK/PD study designs in the macaque model to determine their suitability for the evaluation of drugs to combat tuberculosis. Model substances have been the drugs Isoniazid, Rifampicin, Ethambutol and Pyrazinamide, which have previously been shown to successfully treat both natural and experimental TB infection in macaques and are the standard drugs in the treatment of tuberculosis in man.
The aim of the study is to apply PK/PD models in non-human primates in the development of new anti-TB drug candidates to treat human tuberculosis.
Whereas the study design is optimal as well as the results, there is no discussion whether the model predicts the well-known pharmacokinetics and pharmacodynamics of the 4 standard drugs in humans. Without this the study remains a nice exercise in macaques and does not warrant publication.
This was an exercise in seeking to refine the models used for preclinical testing of new TB drugs by using drugs with proven efficacy to test the new approaches for evaluation. Therefore, the concept was based on the expectation that the four drugs tested are effective against TB in macaques (as demonstrated by Lin et al 2013) as they are in humans and therefore the drug combinations would provide 'positive controls' that would enable the assessment of the utility of the new study designs for efficacy testing. The drug regimens selected for evaluation were designed to enable comparisons of highly effective and less effective treatment regimens to evaluate the sensitivity of the study designs for identifying treatment efficacy.
We also believe that in line with the ARRIVE guidelines for the publication of animal studies it is important that the work is published to demonstrate that the cross over designs for TB drug evaluation is not optimal for macaque studies which could limit this work being repeated and saving further animals being used to pursue this approach
Reviewer 2 Report
Mycobacterium tuberculosis infection is becoming a global issue. With increasing disease prevalence and resistance to standard treatments, the authors here tried to build a non-human primate model for drug Pk/PD studies. The applied animal model is important for new drug development. The author provided the model simulating human studies for dosing levels, sampling times and outcome analysis. Generally, the NHP animal model is helpful for human drug development. However, I have some questions for study designs if clarified could make the article more comprehensive.
1. Introduction section, line 78, what is 3Rs agenda? Please explain in detail.
2. Figure 1 A, please explain the rationale for the treatment protocol. Some studies animal received phase 1 treatment but hold treatment on phase 2, then continued treatment on phase 3 and phase 4. The protocol design was confusing.
3. Section clinical assessment, line 260-269. Why wouldn’t the author monitor liver function and renal function? For liver and renal function also affects Pk/PD of drugs and is related to treatment outcomes.
4. Line 522, most of the animals had pneumonia. Pneumoia is a typing error.
5. The resolution of figure 3 is inadequate. When zoomed in, the details blur.
Author Response
- Introduction section, line 78, what is 3Rs agenda? Please explain in detail.
I have added in a sentence to clarify what the 3R’s are.
The potential to reduce the number of animals required for drug evaluation studies would represent an important contribution to the 3Rs agenda in this area, where the aims are to Reduce the number of animals used for experimentation, Replace animals with alternatives including in vitro assays and modelling and Refinement of experiments to improve animal welfare and reduce suffering.
2.Figure 1 A, please explain the rationale for the treatment protocol. Some studies animal received phase 1 treatment but hold treatment on phase 2, then continued treatment on phase 3 and phase 4. The protocol design was confusing.
Lines 70-79 explain the rationale behind the study design – but in essence it was attempting to use a cross-over design to reduce the impact of individual variation by applying the different treatments sequentially in the same animal but I accept that the design is complex.
- Section clinical assessment, line 260-269. Why wouldn’t the author monitor liver function and renal function? For liver and renal function also affects Pk/PD of drugs and is related to treatment outcomes.
Assessment of PK/PD was limited to blood as the capabilities to monitor liver and renal function were not available for application in this study.
- Line 522, most of the animals had pneumonia. Pneumoia is a typing error.
Thank you for bringing this to our attention and it has been rectified.
- The resolution of figure 3 is inadequate. When zoomed in, the details blur.
All figures will be submitted in high resolution for the final submission.
Reviewer 3 Report
This report describes the application of novel PK and PK/PD study designs in the macaque model to determine their suitability for the evaluation of drugs to combat tuberculosis. Th topic of this paper is interesting, and its writing is good which can be published before minor revisions.
Author Response
This report describes the application of novel PK and PK/PD study designs in the macaque model to determine their suitability for the evaluation of drugs to combat tuberculosis. Th topic of this paper is interesting, and its writing is good which can be published before minor revisions.
Thank you very much for your feedback.
Reviewer 4 Report
-- The study entitled (Determination of the Pharmacokinetics and Pharmacodynamics of Isoniazid, Rifampicin, Pyrazinamide and Ethambutol in a cross-over cynomolgus macaque model of Mycobacterium tuberculosis infection.) is very interesting but the are some inquires about the experiment. - The authors selected 2 strains of M. tb to make challenge, please mention the sensitivity and resistance profile of the selected strains.
- Illustrate the exact time of administration of tested substances after aerosol exposure to M. tb.
- Line 288: For all studies, 2 ml blood samples were collected from each animal immediately prior to the administration of treatment to provide a ‘pre-dose’ PK sample for evaluation of 289 ‘trough’ drug level during the study. Please provide the hematological profile of healthy cynomolgus macaques prior the challenge and administration.
- Line 401: What is the used method to confirm the recovered M. tb. Add reference.
- Line 396: How to handle, prepare, and decontaminate the samples before culturing. Add reference.
- References should be updated.
- Why the author selected IFN-γ ELISPOT method for diagnosis of TB.
- Results should not contain any references as in Lines 451, 466,…etc.
- Results should be more concise
- Provide CT scans, and histopathological findings.
- Lines from 386 to 394 should be transferred to discussion part.
- Give more details about the tested products resistance in examined cynomolgus macaques.
- Give an explanation about using the tested products, without any addition of natural products, or newly tested substances.
- Reference (29): revise it.
- Conclusion part should be added
Author Response
The study entitled (Determination of the Pharmacokinetics and Pharmacodynamics of Isoniazid, Rifampicin, Pyrazinamide and Ethambutol in a cross-over cynomolgus macaque model of Mycobacterium tuberculosis infection.) is very interesting but the are some inquires about the experiment.
The authors selected 2 strains of M. tb to make challenge, please mention the sensitivity and resistance profile of the selected strains.
The H37Rv strain of M. tuberculosis was selected for the first study as this has been shown to be less pathogenic in macaques than the Erdman strain (Gormus 2004) to reduce the risk of animals developing progressive TB-induced disease that met end-point criteria necessitating their removal from study before the planned start of the treatment. This strategy was successful in maintaining all animals in the study for the required time although the highly successful control of H37Rv by the host meant the volume of disease induced was low which reduced the opportunity to identify treatment effects. Consequently, the more virulent TB strain Erdman (Gormus et al 2004) was used for the second study with the aim to increase pulmonary disease burden induced prior to the start of treatment to increase the opportunity to measure treatment effects. The rational for the change in strains is provided in discussion line 808-812
.
Illustrate the exact time of administration of tested substances after aerosol exposure to M. tb.
The start of treatment relative to M.tb challenge for each study are shown in Figure 1, by timelines beneath each treatment plan showing the week after exposure that each phase started. The two diagrams illustrate that in the first study, drug treatment was started 12 weeks following M.tb challenge, whereas in the second study, drug treatment was started 8 weeks after M. tb challenge.
Line 288: For all studies, 2 ml blood samples were collected from each animal immediately prior to the administration of treatment to provide a ‘pre-dose’ PK sample for evaluation of 289 ‘trough’ drug level during the study. Please provide the hematological profile of healthy cynomolgus macaques prior the challenge and administration.
We are unable to provide a haematological profile for the cynomolgus macaques in this study. However information on the haematological profile of cynomolgus macaque of similar age from the same breeding unit can seen in our publication “Differences in host immune populations between rhesus macaques and cynomolgus macaque subspecies in relation to susceptibility to Mycobacterium tuberculosis infection” Sibley et al 2021.
Line 401: What is the used method to confirm the recovered M. tb. Add reference.
Colony were considered as Mtb based on their morphology and growth on selective media the sentence describing the process has been modified for clarity as below:
“Plates were incubated for three weeks at 37ºC, and resultant colonies were counted” as the colonies were confirmed phenotypically as M. tb.
Line 396: How to handle, prepare, and decontaminate the samples before culturing. Add reference.
The samples were not decontaminated prior to plating as they would not have been culturable. All work was carried out in an MSCIII cabinet in a BSL3 laboratory.
ferences should be updated.
References have been updated.
Why the author selected IFN-γ ELISPOT method for diagnosis of TB.
The ELISPOT was applied to monitor the T-cell immune response to infection, and to determine whether antibiotics have an effect on the immune response, as an interaction between drug treatment and the immune system could have a beneficial or detrimental effect on disease progression.
Results should not contain any references as in Lines 451, 466,…etc.
I have removed the references.
Results should be more concise
The work reported spans four PK studies and two PK/PD studies and consequently the volume of data generated was large. Every effort has been made to limit the inclusion of extraneous details.
Provide CT scans, and histopathological findings.
The images derived from the CT scans were evaluated an expert thoracic radiologist highly experienced in the manifestations of TB in macaques, with approaches applied to quantify disease burden to enable interrogation of potential treatment effects. Whilst the radiologist examines all images [slices] of a CT scan, a two-dimensional image can only provide detailed information from a single slice of a CT scan which limits the disease that can be shown. In order to use space efficiently the data presented focuses on the findings from the approaches used to quantify disease burden from the scans and the findings of these key analyses are shown in Figure 3 and 5 and described in the text. The histopathological findings are also described in figure 3 and in the text (line 534-541and 671-675).
Lines from 386 to 394 should be transferred to discussion part.
lines 386-394 refer to how the granulomas are classified and for this reason it is more appropriately described in the methods section than in the discussion.
Give more details about the tested products resistance in examined cynomolgus macaques.
I thank you for this comment and agree that it would be interesting to look at the development of drug resistance in an in vivo model, but that was not part of the remit of this study.
Give an explanation about using the tested products, without any addition of natural products, or newly tested substances.
These studies aimed to refine the models used for the preclinical testing of new drugs by using drugs with proven efficacy to test the new approaches for evaluation. Therefore, the concept was based on the expectation that the four drugs tested which have proven efficacy against human tuberculosis would be effective against TB in macaques (as demonstrated by Lin et al 2013) and therefore the drug combinations would provide 'positive controls' to enable the assessment of new study designs for efficacy testing. The regimens selected were chosen to enable comparisons of highly effective and less effective treatment regimens to evaluate the sensitivity of the study designs for identifying treatment efficacy.
Reference (29): revise it.
The reference has been revised as requested.
Conclusion part should be added
A conclusion is provided in the last paragraph
Round 2
Reviewer 1 Report
No further comments